# Analysis of Factors Influencing Hospitals’ Implementation of a Green E-Procurement System Using a Cloud Model

**DOI:** 10.3390/ijerph16245137

**Published:** 2019-12-16

**Authors:** Hsin-Pin Fu, Tsung-Sheng Chang, Hsiao-Ping Yeh, Yu-Xuan Chen

**Affiliations:** 1Department of Marketing and Distribution, National Kaohsiung University of Science and Technology, Kaohsiung 824, Taiwan; hpyeh2000@nkust.edu.tw (H.-P.Y.); 0626804@nkust.edu.tw (Y.-X.C.); 2Department of Information Management, Da-Yeh University, Changhua 515, Taiwan; china@mail.dyu.edu.tw

**Keywords:** Influencing factors, hospitals, green e-procurement, cloud model, FAHP, TOE

## Abstract

Currently, the green procurement activities of private hospitals in Taiwan follow the self-built green electronic-procurement (e-procurement) system. This requires professional personnel to take the time to regularly update the green specification and software and hardware of the e-procurement system, and the information system maintenance cost is high. In the case of a green e-procurement system crash, the efficiency of green procurement activities for hospitals is affected. If the green e-procurement can be moved to a convenient and trusty cloud computing model, this will enhance the efficiency of procurement activities and reduce the information maintenance cost for private hospitals. However, implementing a cloud model is an issue of technology innovation application and the technology-organization-environment (TOE) framework has been widely applied as the theoretical framework in technology innovation application. In addition, finding the weight of factors is a multi-criteria decision-making (MCDM) issue. Therefore, the present study first collected factors influencing implementation of the cloud mode together with the TOE as the theoretical framework, by reviewing the literature. Therefore, an expert questionnaire was designed and distributed to top managers of 20 private hospitals in southern Taiwan. The fuzzy analysis hierarchical process (FAHP), which is a MCDM tool, finds the weights of the factors influencing private hospitals in southern Taiwan when they implement a cloud green e-procurement system. The research results can enable private hospitals to successfully implement a green e-procurement system through a cloud model by optimizing resource allocation according to the weight of each factor. In addition, the results of this research can help cloud service providers of green e-procurement understand users’ needs and develop relevant cloud solutions and marketing strategies.

## 1. Introduction

Affected by increasingly serious environmental pollution and climate change issues, and global environmental awareness, global economies have enacted relevant environmental protection regulations, requiring enterprises to take responsibility for protecting the environment. The European Union (EU) is the first economic area to propose green procurement standards and the required environmental protection regulations, the Waste Electrical and Electronic Equipment Directive (WEEE) and restriction of the use of certain hazardous substances in electrical and electronic equipment (RoHS). In addition, countries such as the United States and China have also successively formulated environmental protection regulations. Equipment suppliers have formed a certain pressure of restraint and market forces are urging them to establish green production and procurement specifications [1]. Medical devices are also included in the WEEE regulations [2] and these medical equipment suppliers actively establish a green supply chain management (G-SCM) to respond to future green business opportunities. Among the activities of G-SCM, the most important is green procurement activities [3,4]. The content of the procurement operation is to find suppliers in accordance with green specifications, inquiry, and bargaining. This is called green electronic procurement (e-procurement) if the green procurement activities are carried out electronically. 

Health care refers to medical care services provided by industries, and the health care industry can be divided into three categories: hospitals, medical and dental practices, and other human health activities [5]. There were 199 large hospitals with emergency rooms in Taiwan in 2018. In order to solve the lack of performance of public hospitals, most of them have adopted the public-owned and private-run model. Therefore, 90% of the 199 large hospitals belong to private hospitals [6]. In the present study, the 20 large private hospitals with emergency rooms in southern Taiwan are the main research subjects because the procurement activities of private hospitals are more flexible than public hospitals, which are limited by annual budget. In terms of internal management, hospitals must purchase medical goods and equipment to conduct their work and Gabriel et al. [7] indicated that health care goods require a cleaner and safer production environment than general goods, whether they are drugs or equipment. The products used in hospitals are required to have as few side effects as possible on the human body. In addition, the products themselves need to have low energy consumption and decomposable or recyclable characteristics, and product disposal should not create environment pollution. Many hospitals have gradually increased the demand for green procurement and green procurement is becoming increasingly relevant.

The choice of the “right” suppliers and the preservation of an effective interaction with them are important factors in green purchasing activities. Choosing green suppliers requires organizations to first evaluate the benefits and values derived from green procurement [8]. Green procurement creates considerable difficulties, mainly because the materials and technologies required by each supplier differ [9]. Whether suppliers can accommodate the production of green commodities is also of pertinent concern. Although green e-procurement is a type of procurement activity, it requires relatively stricter conditions in the choice of procurement goods and suppliers than other activities. The supplier of medical equipment must meet green commodity standards before they can be listed as suppliers of hospitals and there are certain difficulties to be met to become a green supplier. Hospitals must face the issue of whether the existing green procurement activities can be compatible with their current green e-procurement system.

Currently, the architecture of a green e-procurement system is mostly based on a self-built system within the private hospitals. The client–server model and the database and system are built within the hospitals. This system may be developed by the private hospitals’ information technology staff or set up with the help of external software vendors. The self-built green e-procurement system requires professional personnel to take the time to regularly update the green specifications, software and hardware of the green e-procurement system, and the information system maintenance cost is high. In the case of a green e-procurement system crash, the efficiency of green procurement activities for hospitals is affected. If the green e-procurement system of hospitals can be moved to a convenient and trusty cloud architecture model, the information maintenance cost can be reduced and the efficiency of purchasing activities can be improved for hospitals.

The cloud model consists of three architectures: software as a service (SaaS), platform as a service, and infrastructure as a service [10]. The SaaS architecture is the most popular, mainly because of its inherent benefits: quick setup and deployment, and a low investment cost. These benefits from SaaS reduce the obstacles faced by enterprises in establishing a cloud e-procurement system. The cloud model also offers immediacy and a quick response to users, as well as improved efficiency of purchasing activities, and it helps personnel to understand the status of purchasing raw materials [11,12]. Therefore, more and more enterprises or institutions are investing large resources in their implemented cloud computing service. However, the effect is not as good as expected [13]. However, implementing cloud systems has certain risks, such as the security of data storage and resistance of organization personnel, etc. [13]. If private hospitals understand the weight of factors affecting implementation of the cloud service, their implementation of green e-procurement can then be facilitated and the success rate of cloud model implementation is enhanced. Therefore, the purpose of this paper was to find the weights of factors affecting private hospitals’ implementation of green e-procurement through a cloud model by conducting an expert survey.

Several studies have implemented the technology acceptance model and information systems success model for the implementation of e-procurement in companies [14,15]. In addition, previous research on the implementation of technology innovations has used the technology-organization-environment (TOE) framework, because this architecture clearly defines three main facets when organizations implement innovative technology processes: technology, organization and environment [16]. The technology facet describes the technology related to internal and external organizations, including the innovative technology in the organization and the existing technology available in the market. The organization facet includes the size of the organization, the complexity of the organizational structure, the quality of human resources, and the amount of resources available internally. The environment facet refers to competitors, suppliers, infrastructure and government regulation. To implement a cloud green e-procurement service is a technological innovation for organizations. Therefore, the technology-organization-environment (TOE) framework is a suitable and mature theory that can help private hospitals understand important factors of implementing a green e-procurement system through a cloud model.

Finding the weight of factors affecting the implementation of a cloud green e-procurement system in hospitals is a multi-criteria decision making (MCDM) issue. Therefore, the present study first collected the factors affecting organizations implementing a cloud model from a literature review and constructed a three hierarchical factor table using a TOE framework. The expertise questionnaire was designed and distributed to top information technology managers in large private hospitals in southern Taiwan. Finally, a MCDM tool was used to find the weight of the factors. 

## 2. Literature Review

### 2.1. SCM in Cloud Computing

Lindner et al. [17] compared the cloud suppler chain management (SCM) mode with the self-built SCM mode and discovered differences. First, the implementation cost of the cloud SCM mode is lower than the self-built SCM mode. The database and system server are separated, and a multiclass service level agreement is provided when adopting cloud computing. Second, the system can be used by various users and the computing feedback speed is increased. Hazen et al. [18] also demonstrated that the data processing speed of a cloud SCM system is faster than a conventional SCM system. Overall, SCM using cloud computing yielded more advantages than conventional SCM. Given that companies face competition and pressure from internal and external environmental uncertainties, the properties of cloud computing information processing will encourage companies to implement it [19]. Establishing green procurement specification on SCM for the procurement of green products can be called the green SCM (G-SCM). Among the activities of G-SCM, the most important is green e-procurement.

### 2.2. Green Procurement

Green procurement emphasizes the importance of procurement activities and the quality of the products purchased by enterprises must meet green standards. Generally, if suppliers meet the certification standards of the Leadership in Energy and Environmental Design, they are considered to be green suppliers. However, some companies set their own standards. For example, Walmart proposed the company’s green procurement policy in 2009. Their policy regulates products’ sustainability indices and evaluates the effects of products on the environment. Peattie and Crane [20] also reported that green management influences environmental sustainability. The implementation of green procurement was first promoted in government agencies before being extended to private enterprises. Although both the agency and the supplier agree that purchasing “pure” goods is a correct policy that will increase the threshold of many purchases and maintain a neutral attitude, it still has considerable obstacles in the way of implementation [21]. Zhu et al. [22] reported that external requirements and norms are among the main motivations leading companies to adopt green procurement. In addition, Schaltenbrand et al. [23] advised business owners to carefully evaluate their investment in green management because it does not necessarily lead to substantial profit in the market. This trend is mainly due to the promotion of environmental awareness as part of national policy. Furthermore, Igarashi et al. [24] observed that the main focus in green procurement has gradually shifted from early willingness to implement, forecasting, and planning in the past to supplier selection assessment, processing, performance, and value.

### 2.3. TOE Framework

Depietro et al. [16] proposed the TOE framework, which consists of three factors: technology, organization, and environment. Academically, the TOE framework is a mature theory, and several studies have been conducted on its use in information and communication technology, such as enterprise resource planning [25], a mobile application for booking hotels [26], and considered factors of manufacturing adopting 3D printing [27]. In all of the aforementioned examples, the TOE framework was used as the theoretical basis for the research. The TOE framework has been tested by the aforementioned empirical studies of organizations in various information system domains and innovative technology applications. Implementing a cloud green e-procurement system is also an innovative technology application and TOE is a suitable research framework in the presented study.

Based on the prior description, recent research on the use of a cloud computing service in green procurement in hospitals is limited, making this topic worthy of investigation. The TOE framework takes various relevant factors from different perspectives into account [16]. In the past, most research methods for determining the factors influencing the green supply chain or cloud computing service were based mainly on a regression analysis model focused on the influence of the factors rather than their importance (weight). Although the β value in the regression analysis model can demonstrate the importance of a factor, the estimated β value can be biased or negative [28]. Only a few studies have used the β value from the regression analysis model to determine the weight (or importance) of the factors. Determining the weight (or importance) of factors in cloud computing also concerns multi-criteria decision making (MCDM). Therefore, fuzzy analytical hierarchy process (FAHP), which is a MCDM tool, was used in the present study together with TOE as the theoretical framework to understand the importance of factors influencing private hospitals when they are implementing a green e-procurement system through a cloud model.

## 3. Research Method

Evaluating the weight of the factors can be achieved through MCDM. The analytical hierarchical process (AHP) is the most popular approach [29,30], but it has shortcomings. When applying AHP, the ambiguity of human thinking affects experts’ answers to the questionnaires, their measurement criteria, as well as their subjective judgments [30]. As a result, van Laarhoved and Pedrycz [31] considered the ambiguity of human thinking and introduced FAHP. Because FAHP considers the uncertainty of a problem, multiple criteria, and experts’ opinions, especially in cases where numerous decision criteria and alternatives are available, it can avoid an overly subjective pairwise comparison value and provide a more precise result. Dang et al. [32] adopted it to evaluate several cities in an emerging economy (Vietnam). In the present study, FAHP was used as the research method. The steps to conduct FAHP are as follows.

### 3.1. Identify Assessment Issues

The purpose of this study is to explore the factors considered by private hospitals in southern Taiwan when choosing to use a green e-procurement system with a cloud model and to determine the relative weight of each factor.

### 3.2. Establish a Hierarchical Table

According to the relevant literature reviewed, the criteria influencing the decision-making process were screened and categorized, and the definition of each evaluation criterion was based on the criteria, sub-criteria, and factor layers. In addition, all decision-making factors were expressed in a hierarchical table. To conduct an effective pairwise comparison and achieve favorable consistency, the number of criteria for each layer could not exceed seven.

### 3.3. Design Questionnaires

A questionnaire using FAHP was designed to compare the factors in pairs based on the hierarchical factor layer table. This questionnaire provides clear answers from interviewees, and the collected questionnaires can be analyzed.

### 3.4. Establish Fuzzy Numbers and a Fuzzy Positive Reciprocal Matrix

Fuzzy numbers were established according to the content of the questionnaire. Generally, there are two types of fuzzy numbers, namely triangular and trapezoidal. In the present study, trapezoidal fuzzy numbers were used. The symbols used to represent trapezoidal fuzzy numbers are *a*, *b*, *c*, and *d* (Figure 1). In Figure 1, the fuzzy numbers represented by *a* and *d* are the minimum and maximum value of the membership function, respectively. The fuzzy numbers represented by *b* and *c* are the smallest average and largest average value of the interval, respectively. A fuzzy positive reciprocal matrix was constructed by using the trapezoidal fuzzy numbers previously proposed. When there are *n* sub-criteria in the same criterion, the *n* × *n* fuzzy positive reciprocal matrix *A* can be constructed. 

Where $y(x)=\left\{ \begin{array}{l} \;\;0\;,\;\;\;x<a\\ \frac{x-a}{b-a}\;,\;\;a\leq x<b\\ \;\;1\;,\;\;b\leq x\leq c\\ \frac{d-x}{d-c}\;,\;\;c<x\leq d\\ \;\;0\;,\;\;x>d \end{array}\right.$

### 3.5. Consistency Check

Before performing the weighting operation, the fuzzy weight of the positive reciprocal matrix is used to find the fuzzy number by using the geometric mean, and the consistency check is performed, including the consistency index (CI) and the consistency ratio (CR). 

The function can be written as *CI* = (λ_max_ − *n*)/(*n* − 1), *CR* = (*CI*/*RI_n_*) × 100%, where λ_max_ is the largest eigenvector of the pairwise comparison matrix, *n* presents the number of factors and *RI_n_* is the randomization index [30,31]. The values are presented in Table 1. Saaty [30] suggested that a *CR* value of ≤0.1 is the acceptable range.

### 3.6. Establish a Starting Matrix and Perform Defuzzification

After the consistency check, the fuzzy positive reciprocal matrix was used to establish a fuzzy starting matrix. The present study followed the defuzzification method of α-cut (Lambda–Max method) proposed by Csutora and Buckley [33] to determine the value of the fuzzy weight interval (*W*_01_***, *W*_1*l*_***, *W*_1*u*_***, *W*_0*u*_***). The Lambda–Max method has the following three advantages: (1) it can accommodate any fuzzy pairwise comparison type, (2) it is easy to compute without calculation and only positive matrix-vector and eigenvector values are needed, and (3) it reduces the fuzziness compared with other methods. When performing α-cut in the positive reciprocal matrix, the *A_αl_* and *A_αu_* matrix are formed first, where *A_αl_* is the *b* value of the fuzzy number and *A_αu_* is the *c* value of the fuzzy number. Subsequently, the *n* eigenvalues of the matrix *A_αl_* and *A_αu_* were obtained and normalized. Afterwards, *W*_1*l*_ = (*W_1l1_*,…,*W*_1*ln*_), *W*_1*u*_ = (*W*_1*ul*_,…,*W*_1*un*_), and the fuzzy weight interval (*W*_01_***, *W*_1*l*_***, *W*_1*u*_***, *W*_0*u*_***) were obtained using α-cut procedures. The geometric mean was subsequently used to calculate the fuzzy weight interval value to be the explicit weight value (*W**). The detailed calculation α-cut procedures can be found in Csutora and Buckley [33]. 

### 3.7. Normalization

Finally, the weight values were normalized to obtain the local weights in three layers. On the basis of this information, the global weights for all factors were calculated.

## 4. Data Collection

Following the FAHP, the weight of the factors considered by hospitals when implementing green e-procurement systems through a cloud model were determined based on the TOE framework. In the present study, a TOE hierarchical table of the factors when implementing the cloud computing service [34] was first established. A three-layer hierarchical factor table based on the TOE framework included a criteria layer, a sub-criteria layer, and a factor layer. Basically, TOE in the criteria layer is the main framework. The technology facet mainly included system security, system quality, and system function in the sub-criteria layer [34]. The organization facet covered organizational support, organizational characteristics, and organizational readiness in the sub-criteria layer [34]. The environment facet also covered industrial environment, overall environment, and cloud service providers in the sub-criteria layer [34].

In the factors layer, 27 factors were collected as following. Hassan et al. [18] mentioned that senior management, employee acceptance, interdepartmental coordination, organizational systems, organizational infrastructure, usable resources, degree of industrial adoption, pressure of market competition and reasonable charge of the cloud service influenced cloud computing adoption in small and medium enterprises. In addition, Amiri [35] also pointed out that the degree of education training and data access security were factors adopting SaaS. Furthermore, Arpaci [36] mentioned that data access security, information transmission security, usefulness of system operations and innovation and design were factors of cloud computing adoption in education that achieve knowledge management. Raut et al. [37] mentioned that data access security, information system integration, information reliability and system expandability were the critical success factors of cloud computing adoption in the micro, small and medium enterprises (MSMEs). Priyadarshinee et al. [38] verified that system security, including data access security, information transmission security and fallback cloud management security, were the determinants of cloud computing adoption. Ke et al. [39] also indicated that information systems and communication stability could influence the cloud computing platform performance. In addition, Vázquez-Poletti et al. [40] demonstrated that information system integration and infrastructure influenced the SaaS operation in a cloud computing environment. In addition, Arpaci [41] pointed out that ease of use of the system operation influenced students’ intention to use mobile cloud storage services. Chang [42] proposed a framework for cloud computing adoption, covering senior management, ability of cloud service providers and relationship with cloud service providers. Safari et al. [43] pointed out that interdepartmental coordination and pressure of market competition were the determinants of the adoption of SaaS. Kim et al. [44] verified that degree of industrial adoption and promotion of government policy were the determinants of software-as-a-service adoption in small businesses. Moreno-Vozmediano et al. [45] pointed out that development of cloud service industry was a key challenge in cloud computing adoption. Asadi et al. [46] verified that reasonable charge of cloud service and relationship with cloud service providers also influenced adoption of cloud computing in the banking sector. The aforementioned studies focused on factors or determinants of adopting the cloud model service. Finally, summarizing the factors affecting the adoption of cloud service models mentioned in the above studies, a three-layer hierarchical table containing 27 factors is shown in Table 2. 

The pairwise questionnaires in this study were designed according to the suggestions of Saaty and Vargas [47] after the hierarchical factor table was established. The factors in each layer were measured using a ratio scale and a pairwise comparison method. Their relative importance was compared, and then their importance was ranked in an ascending order and expressed by 1, 2, 3… 9. For instance, Table 3 is a questionnaire of the criteria layer comparing the technology facet, organization facet and environment facet. The factors were measured using a ratio scale. Their relative importance was compared in ascending order from 1 to 9. Based on the criteria layer, the respondents selected the relative importance of 2 factors out of 3 (technology, organization, and environment), as presented in Table 3. As shown in Table 3, the respondents appeared to think that technology is four times more important than organization and two times more important than environment, and that environment is two times more important than organization. The questionnaire design of the other layers is the same as that of the criteria layer.

Once the questionnaire was finalized, it was distributed to the top information technology managers at 20 large private hospitals in southern Taiwan. Because the topic of this research was the implementation of a cloud model by an experts’ survey, it was crucial to consider whether the experts were representative. In other words, the experts answering the questionnaires were required to have a long-term career and in-depth understanding in the field [48]. That is, questionnaires answered by experts from this field are more representative of field personnel when implementing a cloud service model and yield results that are closer to the actual situation of the field. Therefore, the experts selected in the present study were senior information technology managers from the southern Taiwanese private hospitals who had been working for at least 5 years. After the questionnaires were collected and reviewed, all appeared to have passed the consistency check. Because FAHP is a hierarchical analysis method and not a regression analysis method, the study did not require a large sample size [49]. With regard to the sample size for expert questionnaires, Delbecq et al. [50] claimed that 15 to 30 participants was a reasonable sample size, while Robbins [51] stated that 5 to 7 participants were sufficient if the group of experts was highly homogeneous. The 20 participants to answer the expert questionnaire in the current study were representative for this research topic. Although only 20 valid expert questionnaires were obtained, the number of samples required and the experts’ background and experience were in line with the research method. The samples could be used for research analysis. Finally, analysis of the questionnaire data was conducted following FAHP. Assume Table 3 is the response of the first expert, the weight calculation was performed as follows. 

### 4.1. Starting Matrix Establishment

Because *a*_12_ = 4, *a*_13_ = 2, *a*_23_ = 1/2, based on the response of the first expert in Table 3, the following is the starting matrix formula:$$A=\left[a_{ij}\right]=\left[\begin{array}{cc} \begin{array}{cc} 1 & a_{12}\\ 1/a_{12} & 1 \end{array} & \begin{array}{cc} \Lambda & a_{1n}\\ \Lambda & a_{2n} \end{array}\\ \begin{array}{cc} Μ & Μ\\ 1/a_{1n} & 1/a_{2n} \end{array} & \begin{array}{cc} O & Μ\\ \Lambda & 1 \end{array} \end{array}\right]=\left[\begin{array}{ccc} 1 & 4 & 2\\ \frac{1}{4} & 1 & \frac{1}{2}\\ \frac{1}{2} & 2 & 1 \end{array}\right]$$

### 4.2. Consistency Check

*CI* and *CR* values were calculated as follows:$$\left[\begin{array}{ccc} 1 & 4 & 2\\ \frac{1}{4} & 1 & \frac{1}{2}\\ \frac{1}{2} & 2 & 1 \end{array}\right]\times \left[\begin{array}{l} W_{1}\\ W_{2}\\ W_{3} \end{array}\right]=\left[\begin{array}{l} {W^{\prime }}_{1}\\ {W^{\prime }}_{2}\\ {W^{\prime }}_{3} \end{array}\right],$$
$$\lambda_{max}=\;\frac{1}{n}\left(\frac{{W^{\prime }}_{1}}{W_{1}}+\frac{{W^{\prime }}_{2}}{W_{2}}+\Lambda +\frac{{W^{\prime }}_{n}}{W_{n}}\right)$$
$$W_{i}={\left[\overset{n}{\underset{j=1}{\prod }}aij\right]}^{\frac{1}{n}}/\overset{n}{\underset{i=0}{\sum }}{\left[\overset{n}{\underset{j=1}{\prod }}aij\right]}^{\frac{1}{n}}$$

*i*, *j* = 1, 2, 3…, *n*. (There were three factors, so *n* = 3.)

From this formula, we calculated *W_i_* as follows: *W*_1_ = 4/7 = 0.571429, *W*_2_ = 1/7 = 0.142857, *W*_3_ = 2/7 = 0.285714. *W*′_1_ was then calculated as follows: *W*′_1_ = (1 × 571429) + (4 × 0.142857) + (2 × 0.285714) = 1.714285, *W*′_2_ = 0.428571 and *W*′_3_ = 0.857143, so $\lambda_{max}$ = 1/3 (1.714285/0.571429 + 0.428571/0.142857 + 0.857143/0.285714) = 3.000001.
$$\begin{array}{l} CI\;=\;(\lambda_{max}-n)/\left(n-1\right)\;=\;(3.000001\;-\;3)/(3\;-\;1)\;=\;0.002002,\\ CR=\left(CI/RI_{n}\right)\;=\;0.002002/0.525\;=\;0.003813. \end{array}$$

Both *CI* and *CR* values were less than 0.1. The values met the consistency requirement and were acceptable.

### 4.3. Fuzzy Numbers and Fuzzy Positive Reciprocal Matrices were Constructed

According to Table 3, the trapezoidal fuzzy number can be displayed as: [3,4,4,5], [1,2,2,3], and [1/3,1/2,1/2,1].

A fuzzy positive reciprocal matrix is established as follows:$$A=\left[\begin{array}{c} \;\begin{array}{ccc} A11\; & A12 & \;A13 \end{array}\\ \begin{array}{ccc} A11\;1,1,1,1 & 3,4,4,5 & 1,2,2,3\\ \;A12\;\frac{1}{5},\frac{1}{4},\frac{1}{4},\frac{1}{3} & 1,1,1,1 & \frac{1}{3},\frac{1}{2},\frac{1}{2},1\\ \;A13\;\frac{1}{3},\frac{1}{2},\frac{1}{2},1 & 1,2,2,3 & 1,1,1,1 \end{array} \end{array}\right]$$

### 4.4. Defuzzying the Fuzzy Number

The α-cut [34] is used to defuzzy the fuzzy matrix and obtain the weights of factor as follows.

Let α = 1 

*A*_1_*_l_*$a_{12}$ = ((b)-(a)) × 1+ (a) = (4-3) × 1 + 3 = 4,

*A*_1_*_u_*$a_{12}$ = (d)-((d)-(c)) × 1 = 5-(5-4) × 1 = 4,

*A*_1_*_l_*$a_{13}$ = ((b)-(a)) × 1+ (a) = (2-1) × 1 + 1 = 2,

*A*_1_*_u_*$a_{13}$ = (d)-((d)-(c)) × 1 = 3-(3-2) × 1 = 2,

*A*_1_*_l_*$a_{23}$ = ((b)-(a)) × 1+ (a) = (1/2-1/4) × 1 + 1/4 = 1/2,

*A*_1_*_u_*$a_{23}$ = (d)-((d)-(c)) × 1 = 1/2-(1/2-1/2) × 1 = 1/2

The fuzzy positive matrices *A*_1 *l*_ and *A*_1*u*_ are displayed as follows:$$A_{1l}=\left[\begin{array}{ccc} 1 & 4 & 2\\ \frac{1}{4} & 1 & \frac{1}{2}\\ \frac{1}{2} & 2 & 1 \end{array}\right],\;A_{1u}=\left[\begin{array}{ccc} 1 & 4 & 2\\ \frac{1}{4} & 1 & \frac{1}{2}\\ \frac{1}{2} & 2 & 1 \end{array}\right]\;\mathrm{and}\;A_{1\;l}=\;b,\;A1_{u}=\;c$$

*W*_1_, *W*_1*l*_, and *W*_1*u*_ were calculated as follows:

*W*_1*l*_ = (0.571429, 0.142857, 0.285714), *W*_1*u*_ = (0.571429, 0.142857, 0.285714)

$K_{1l}=min$$\left\{\frac{W_{im}}{W_{1il}}|1\leq i\leq n\right\}$, *K*_1*u*_ = max $\left\{\frac{W_{im}}{W_{1iu}}|1\leq i\leq n\right\}$, and *W*_1*m*_ = *W*_1_, *W*_2*m*_ = *W*_2_, *W*_3*m*_ = *W*_3_

*K*_1_*_l_* =min (0.571429/0.571429, 0.142857/0.142857, 0.285714/0.285714) = 1

*K*_1_*_u_* =max (0.571429/0.571429, 0.142857/0.142857, 0.285714/0.285714) = 1

*W**_1*l*_ = *K*_1*l*_ × *W*_1*l*_, *W**_1*u*_ = *K*_1*u*_ × *W*_1*u*_

*W**_1*l*_ = (0.571429, 0.142857, 0.285714), *W**_1*u*_ = (0.571429, 0.142857, 0.285714)

Let α = 0.5. The same calculation sequence of α = 1 is performed again as follows:

*A*_0.5*l*_$a_{12}$ = ((b)-(α))× 0.5 +(a) = (4-3) × 0.5 + 3 = 7/2, 

*A*_0.5*u*_$a_{12}$ =(d)-((d)-(c)) × 1 = 5-(5-4) × 0.5 = 9/2,

*A*_0.5*l*_$a_{13}\;$= ((b)-(a))× 0.5 +(a) = (2-1) × 0.5 + 1 = 3/2, 

*A*_0.5*u*_$a_{13}\;$=(d)-((d)-(c)) × 1 = 3-(3-2) × 0.5 = 5/2,

*A*_0.5*l*_$a_{23}\;$= ((b)-(a))× 0.5 +(a) = (1/2-1/3) × 0.5 + 1/3 = 5/12, 

*A*_0.5*u*_$a_{23}\;$=(d)-((d)-(c)) × 1 = 1-(1-1/2) × 0.5 = 3/4

*A*_0.5*l*_ and *A*_0.5*u*_ are displayed as follows: $$A_{0.5l}=\left[\begin{array}{ccc} 1 & \frac{7}{2} & \frac{3}{2}\\ \frac{2}{7} & 1 & \frac{5}{12}\\ \frac{2}{3} & \frac{12}{5} & 1 \end{array}\right],\;A_{0.5u}=\left[\begin{array}{ccc} 1 & \frac{9}{2} & \frac{5}{2}\\ \frac{2}{9} & 1 & \frac{3}{4}\\ \frac{2}{5} & \frac{4}{3} & 1 \end{array}\right]$$

*W*_0.5*l*_*=* (0.511247, 0.144705, 0.344047), *W*_0.5*u*_ = (0.622075, 0.152783, 0.225143)

*K*_0.5*l*_ = min (0.571429/0.511247, 0.142857/0.14705, 0.285714/0.344047) = 0.830451

*K*_0.5*u*_ = max (0.571429/0.622075, 0.142857/0.152783, 0.285714/0.225143) = 1.269037

*W**_0.5*l*_ = (0.424566, 0.120171, 0.285714), *W**_0.5*u*_ = (0.789435, 0.193887, 0.285714),

Let α = 0,

*A*_0*l*_$a_{12}$ = ((b)-(a))× 0 +(a) = (4-3) × 0 + 3 = 3, 

*A*_0*u*_$a_{12}$ = (d)-((d)-(c)) × 0 = 5-(5-4) × 0 = 5,

*A*_0*l*_$a_{13}\;$= ((b)-(a))× 0 +(a) = (2-1) × 0 + 1 = 1, 

*A*_0*u*_$a_{13}\;$= (d)-((d)-(c)) × 0 = 3-(3-2) × 0 = 3,

*A*_0*l*_$a_{23}\;$= ((b)-(a))× 0 +(a) = (1/2-1/3) × 0 + 1/3 = 1/3, 

*A*_0*u*_$a_{23}\;$= (d)-((d)-(c)) × 0 = 1-(1-1/2) × 0 = 1

*A*_0*l*_ and *A*_0*u*_ are displayed as follows: $$A_{0l}=\left[\begin{array}{ccc} 1 & 3 & 1\\ \frac{1}{3} & 1 & \frac{1}{3}\\ 1 & 3 & 1 \end{array}\right],\;A_{0u}=\left[\begin{array}{ccc} 1 & 5 & 3\\ \frac{1}{5} & 1 & 1\\ \frac{1}{3} & 1 & 1 \end{array}\right]$$

*W*_0*l*_ = (0.428571, 0.142857, 0.428571), *W*_0*u*_ = (0.658644, 0.156182, 0.185174),

*K*_0*l*_ = min (0.424566/0.428571, 0.120171/0.142857, 0.285714/0. 428571) = 0.666667

*K*_0*u*_ = max (0.789435/0.658644, 0.193887/0.156182, 0.285714/0.185174) = 1.54295

*W**_0*l*_ = (0.285714, 0.095238, 0.285714), *W**_0*u*_ = (1.016255, 0.240981, 0.285714),

After obtaining the fuzzy interval weights (*W**_01_, *W**_1*l*_, *W**_1*u*_, and *W**_0*u*_), the arithmetic mean method is used to obtain the weight of technology factor (*W**):

The fuzzy interval weights (*W**_01_, *W**_1*l*_, *W**_1*u*_, and *W**_0*u*_) of the technology factor are 0.285714, 0.571429, 0.571429, and 1.01625, so the weight of the technology factor is 0.611207, and the weights of the organization factor and environment factor are 0.155483 and 0.285714, respectively.

### 4.5. Normalization

The normalization of *W** from the technology factor, organization factor, and environment factor was normalized as [0.611207/(0.611207 + 0.155483 + 0.285714), 0.155483/(0.611207 + 0.155483 + 0.285714), 0.285714/(0.611207 + 0.155483 + 0.285714)] = (0.580772, 0.147741, 0.271487).

After the weights of the technology, organization, and environment factors in the first respondent’s questionnaire were obtained, the weights of the technology, organization, and environment factors from the second to twentieth expert questionnaires were obtained through the same procedure. The final local weights of the technology, organization, and environment factors were obtained with average values of the 20 expert questionnaires, and the local weight of each factor was obtained through the same procedure. Therefore, all local factor weights (columns A, B and C) across different hierarchies were connected in a series to obtain the global weights of all factors (column C = B × A and column E = D × C) (Table 4).

## 5. Result and Analysis

Table 4 shows the factors by order of importance. The most important factor for private hospitals in southern Taiwan is support from senior management. Due to the implementation of new technology, members of organizations could be unfamiliar and cumbersome, and would therefore be quite resistant. Therefore, this must have the support from senior management so that the senior manager can help cross-departmental communication to gain cross-departmental recognition and cooperation. Therefore, senior managers should have some understanding of the advantages brought by the green e-procurement system through the cloud model. Hales [52] reported that for senior managers to support their subordinates’ missions, they must have an understanding of what challenges they will be facing and how vast the task is. In addition, De Brún and McAuliffe [53] indicated that senior managers require more social skills and the ability to articulate organization’s goals in order to lead the team. In summary, support from senior management in private hospitals in southern Taiwan is necessary, and senior managers should have a basic understanding of innovative information technology applications in the field before they give their support to the implementation of a cloud model.

The second most important factor private hospitals are concerned with in southern Taiwan is data access security. In general, the cloud architecture of a supply chain procurement system is semi-open or open to enable suppliers to access data from the system instantly. Therefore, the security of data access is crucial. From a technical perspective, the cloud architecture currently available provides a secure encryption mechanism that ensures data security when personnel log in to the system. With the evolution of information technology in terms of cloud architecture over the past years, users are now able to benefit from a higher security encryption mechanism in a shorter time [54]. However, from a user perspective, security concerns still exist related to accessing data from a cloud computing system, especially in the Internet context where people have often heard about accounts being hacked. Therefore, the cloud system includes recording and displaying functions for every access detail of a person’s accounts to increase user trust in the system.

Information system and communication stability is the third most important factor to be considered. It belongs to the system quality of the sub-criterion layer. Among all factors, information reliability and information system integration are respectively the fifth and sixth factors to be considered. Therefore, we can infer that system quality is regarded as the most important sub-criterion as it accounts for 20.57% of the factor weight. System quality is one of the important variables of a successful cloud green e-procurement model. It directly affects a private hospital’s willingness to use, and their satisfaction with, a cloud model. Therefore, those factors related to system quality cannot be ignored. That is, the implementation of green e-procurement systems through a cloud model requires a stable communication environment. Moreover, this has to be able to integrate the systems already existing in a hospital to ensure consistency in the data and coexistence between the program functions to provide a reliable environment for private hospitals.

The fourth most important factor considered is the degree of industrial adoption. This factor is also the first most important among the environmental facets. Although some private hospitals appear profitable, the managers of a hospital should still be economical to avoid excess resource wastage. Therefore, private hospitals’ management must consider the field-related condition of the external environment and whether other private hospitals are implementing the same strategy or have already done so. If management of a private hospital realizes that others are widely using a cloud green e-procurement system, they are likely to implement it as well.

The factors ranked 7th to 9th of the system function of the sub-criterion layer account for 15.55% of the factor weight. The primary factor in the system function is the usefulness of system operations. In other words, the green cloud e-procurement system must be able to complete executive tasks. The following factor is the ease of use of the operation of the system. The system must be easy to operate and able to send immediate feedback to users for them to quickly become familiar with the cloud model. The 9th factor is system expandability. In the past, most private hospitals’ systems were built on an IBM Unix architecture in Taiwan, the main feature of which was to provide a highly stable environment but with less expandability. The current hospital information system architecture is more diverse. In addition to the C++ and JAVA programming languages, Web service knowledge is required. Therefore, a system’s expandability is more important than it was before. This phenomenon is due to technology driving market demand.

In addition, according to the results in Table 4, the weight of the technological criteria is 53.91% of the total, which is far beyond the weight of the organizational and environmental facets. This indicates that the current southern Taiwan private hospitals highly value the technological facet. This context provides a good opportunity for cloud computing service providers to market and promote their services. However, some hidden concerns exist. Because of the substantial amount of attention being paid to the technological facet, two myths could exist related to this facet. One is that the private hospital has high expectations and is dependent on technology; therefore, hospitals might consider that their problems can be solved by implementing the cloud model and thus ignore the corresponding risks. The other is that the private hospital still lacks the confidence to implement cloud model services but hopes that it will provide them with greater security and a better system integration capability. Therefore, cloud service providers should attempt to prevent the private hospital from succumbing to the aforementioned myths. The main reason for pushing the private hospital to implement a cloud model is rooted in the technological facet. In other words, the capability and security provided by cloud service providers will drive the private hospitals’ implementation of the cloud model.

In summary, the support of senior management has a great impact on the implementation of cloud green e-procurement and senior managers of private hospitals have to understand the situation of implementing cloud green e-procurement by private hospitals in southern Taiwan. The cloud service provider should design the most effective solution in accordance with clients’ need, such as data access security, information systems, communication stability and system function (including usefulness of system operations, system expandability and ease of use of the system operation). In addition to providing the trust, stability and high service level agreement (SLA) of the cloud green e-procurement model, the cloud service providers must have the ability to master all kinds of green regulations and assist with immediate updates. If the cloud service provider can provide the best choice for the senior management of private hospitals in southern Taiwan with the most effective solution and the senior managers of private hospitals agree with the vendors’ solutions, then the hospital is more likely to implement the cloud green e-procurement system and the chances of success in implementing the cloud system under the support of senior management will be higher. On the contrary, if the support of senior management is weak, this may result in failure of the implementation of innovative technology.

The present study only included large hospitals in southern Taiwan. Although the TOE framework and the MCDM tool were used to support the results in this paper, the outcomes might not be in line with other area and cultural settings. Different theoretical backgrounds with a dissimilar research subject, research method and expert panel might yield different research results.

## 6. Conclusions

In the present study, an experts’ survey was conducted among southern Taiwanese private hospital top management of the IT department to understand the importance of factors influencing the implementation of green e-procurement systems through the cloud model. Similar studies in the past have used regression analysis as the research method. Because finding weights of factors is also a MCDM method, a MCDM method (FAHP) was then used to find weights of factors in the present paper. This paper finds that the first five important factors influencing private hospital implementation of cloud green e-procurement system are: support from senior management, data access security, information system and communication stability, the degree of industrial adoption and system function (usefulness of system operations, system expandability and ease of use of the system operation). If private hospitals in southern Taiwan understand the weight of important factors affecting the implementation of a cloud model, this can enable private hospitals to implement a green e-procurement system through a cloud model by optimizing resource allocation according to the weight of each factor. In addition, if cloud computing service providers develop a cloud green e-procurement system based on clients’ need for the senior management of private hospitals in southern Taiwan, the private hospitals are more likely to implement the cloud green e-procurement system. Finally, this study is one of a few that used FAHP to investigate green e-procurement through a cloud model for private hospitals, which is a topic worth studying.

## Figures and Tables

**Figure 1 ijerph-16-05137-f001:**
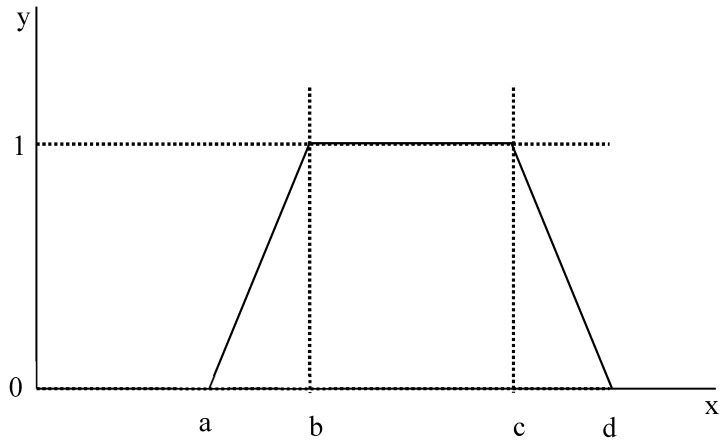
Trapezoidal form of fuzzy numbers.

**Table 1 ijerph-16-05137-t001:** Randomized index of randomization index (*RI_n_*)

*n*	3	4	5	6	7	8	9	10	11	12	13	14	15	16
*RI_n_*	0.525	0.882	1.115	1.252	1.341	1.404	1.452	1.484	1.513	1.535	1.555	1.570	1.583	1.595

**Table 2 ijerph-16-05137-t002:** Three-layer hierarchical factors.

Criteria	Sub-Criteria	Factors
Technology	System security [34]	Data access security [35,36,37,38]
		Information transmission security [36,38]
		Fallback cloud management security [39]
	System quality [34]	Information system and communication stability [39]
		Information system integration [37,39,40]
		Information reliability [37]
	System function [34]	Usefulness of system operations [36]
		System expandability [37]
		Ease to use of system operation [41]
Organization	Organizational support [34]	Senior management [18,42]
		Employee acceptance [18,42]
		Interdepartmental coordination [18,43]
	Organizational characteristics [34]	Organization scale [43]
		Innovative and design ability of organizational process [36,38]
		Organizational system [18]
	Organizational readiness [34]	Organizational infrastructure [18]
		Degree of education training [35]
		Usable resource [18]
Environment	Industrial environment [34]	Degree of industrial adoption [18,44]
		Development of cloud service industry [45]
		Pressure of market competition [18,43]
	Overall environment [34]	Promotion of Government policy [44]
		Government regulation [44]
		National infrastructure [40]
	Cloud service providers [34]	Reasonable charge of cloud service [18,46]
		Ability of cloud service providers [42]
		Relationship with cloud service providers [42,46]

**Table 3 ijerph-16-05137-t003:** Examples of questionnaires.

Factor	9	8	7	6	5	4	3	2	1	1/2	1/3	1/4	1/5	1/6	1/7	1/8	1/9	Factor
Technology	□	□	□	□	□	v	□	□	□	□	□	□	□	□	□	□	□	Organization
Technology	□	□	□	□	□	□	□	v	□	□	□	□	□	□	□	□	□	Environment
Organization	□	□	□	□	□	□	□	□	□	v	□	□	□	□	□	□	□	Environment

Note: comparing the importance of technology, organization and environment and tick where appropriate.

**Table 4 ijerph-16-05137-t004:** Weights of the three-layer hierarchical factors.

Criteria	Weights (A)	Sub-Criteria	Local Weights (B)	Global Weights (C)	Factors	Local Weights (D)	Global Weights (E) [Ranking]
Technology	0.5391	System security	0.3300	0.1779	Data access security	0.5628	0.1001[02]
					Information transmission security	0.2205	0.0392[10]
					Fallback cloud management security	0.2167	0.0385[11]
		System quality	0.3816	0.2057	Information system and communication stability	0.4372	0.0899[03]
					Information system integration	0.2730	0.0561[06]
					Information reliability	0.2898	0.0596[05]
		System function	0.2885	0.1555	Usefulness of system operations	0.3520	0.0547[07]
					System expandability	0.3045	0.0474[09]
					Ease to use of system operation	0.3435	0.0534[08]
Organization	0.2396	Organizational support	0.6676	0.1600	Senior management	0.6283	0.1005[01]
					Employee acceptance	0.2287	0.0366[12]
					Interdepartmental coordination	0.1430	0.0229[17]
		Organizational characteristics	0.1728	0.0414	Organization scale	0.1265	0.0052[26]
					Innovative and design ability of organizational process	0.4162	0.0172[20]
					Organizational system	0.4573	0.0189[18]
		Organizational readiness	0.1596	0.0382	Organizational infrastructure	0.4601	0.0176[19]
					Degree of education training	0.2799	0.0107[21]
					Usable resource	0.2600	0.0099[22]
Environment	0.2213	Industrial environment	0.6273	0.1388	Degree of industrial adoption	0.5560	0.0772[04]
					Development of cloud service industry	0.2107	0.0293[14]
					Pressure of market competition	0.2333	0.0324[13]
		Overall environment	0.1902	0.0421	Promotion of government policy	0.5727	0.0241[16]
					Government regulation	0.2082	0.0088[24]
					National infrastructure	0.2191	0.0092[23]
		Cloud service providers	0.1825	0.0404	Reasonable charge of cloud service	0.7105	0.0287[15]
					Ability of cloud service providers	0.1606	0.0065[25]
					Relationship with cloud service providers	0.1289	0.0052[26]

**Note:** C = B × A, E = D × C. All factors passed the consistency check.

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
