# Peer review of "Analysis of Factors Influencing Hospitals’ Implementation of a Green E-Procurement System Using a Cloud Model"

_ijerph, 2019, doi:10.3390/ijerph16245137_

Round 1

Reviewer 1 Report

The paper is about hospitals' implementation of green a e-procurement system using a cloud model.

The paper has been rewritten, and are now ready for review.

In the abstract a problem appears (row 18), because the authors state that implementing a cloud model has certain risks. This statement is then repeated in the introduction, without telling what the risks are about at all (row 89).

It is difficult to understand where the used sub-criteria and factors (table 2) come from. Is it clear that the sub-criteria and factors are derived from the TOE theory, and that these have a significant effect in previous studies. But which are these studies that have demonstrated the factors significance? If it is the reference related to each of the factor in table 2, this could be clarified in chapter 4, in rows 252-255. Is it the references used in table that constitute the previous studies referred to on row 255?

In the abstract the TOE framework should be presented before the weight of factors, otherwise it is very difficult to related the weight of any factors that is not told about. Try to tell about a defined range of factors, referring to the TOE framework, that have proved to be significant in related studies. Then you can present the understanding of the weight of the factors.....

Rows 40-42, and 44-45 needs references.

On row 50 you state: "the private hospital in Taiwan". Is there only one private hospital in Taiwan. In other places in the paper it seems that there are more than one private hospital, as you refer to private hospitals. It should be fruitful to mention hos many private hospitals there are in Taiwan.

There are now not an appropriate coupling to the text started in row 71. The information about the cloud model is presented quite abruptly, and there is no connection to the previous paragraphs. Try to make a smoother transition to cloud services. You could also reflect about if the rows 71-78 are needed at all.

Check the wording on row 100: "technology described the technology related to...."

You should not present the method or results in the introduction.

Check on row 123-124, cloud Suppler.... conventional SCM and cloud SCM....transitional SCM. It seem that you compare different kinds of SCM, cloud, conventional and transitional....and this is difficult to understand. You may explain this more thoroughly.

In row 131-134 you mention a few factors. You need to mention and motive the factors somewhere in the introduction, before you can refer to them in the text. In the first paragraph in chapter 4 you introduce the factors, and this introduction should be presented in any way in the introduction.

Why is it relevant that German consumers were more interested in purchasing green goods than US citizen? Is this in any way related to hospitals or healthcare goods, or implementation of cloud systems?

A reference is needed for the rows 164-167. The same accounts for the rows 169-170.

Unfortunately, I could not understand the arrangements described on row 260-263, and table 3. Can you please describe this in a more clear way?

How many hospitals do the questionnaires be distributed to?

It is needed that you analyse and discuss your results, at least in relation to the references you have used in table 2, for each of the factors. This is missing for the most of the factors analyzed.

It is unnecessary to open up the limitations to other national and cultural settings in the conclusion. It should be enough to discuss limitations in the context studied. Limitations could be mentioned and discussed in chapter 5, but should not be elaborated on in the conclusions.

As far as I can understand this paper is not discussing or exploring BPR at all. Therefore it is unnecessary to mention such issues in the conclusion. If you think that further studies can be worthful, those suggestions should be related to the results of this study.

Reviewer 2 Report

This manuscript is well-presented and most comments have been addressed.

Major issues:
1. Line 292: the starting matrix A. How did aij come from?
Did it come from the later calculation: lines 308-320? If this is the case, should the order of the paragraph be adjusted?

2. Line 299: How did Wi come from?
Based on the formula of line 297, since a11 = 1, a12 = 4, a13 = 2, a21 = 1/4, a22 = 1, a23 = 1/2, a31 = 1/2, a32 = 2, a33 = 1, we haveW1 = 4/7 = 0.571429, W2 = 1/7 = 0.142857, W3 = 2/7 = 0.285714, which are not the same as line 299: W1 = 0.580772, W2 = 0.147741, W3 = 0.271487.

Minor issues:
1. Line 225-226: is λmax the largest "eigenvalue" of the pairwise comparison matrix?

2. Lines 234 & 240: (W01*, W1l*, W1u*, W0u*) -> the first one should be W0l*, but not W01*?

3. Line 301-302: lamda_max = 1/3 (1.71471/0.580772 + 0.42867/0.147741 + 0.857355/"0.271487") = 3.004003.

Reviewer 3 Report

211 to 219 - these are very basic materials. please put them in an appendix and dont break the flow.

286 - 372 - i suspect that these are called for by the other reviewer. Unfortunately, i cannot see the other reviewer's comments so i am not sure of the context. Nevertheless, i am also suggesting that these calculations be brought to the appendix. just summarize the main calculations in the text. 

Conclusions should be further strengthened.

I have no further comments beyond the pointers above. I am satisfied with the responses to my earlier queries. 

Round 2

Reviewer 1 Report

You will need a small correction in line 22, in the Abstract.

This version of the paper is easy to read and understand, and the text flows logically.

However, there are some work to be done in the Conclusions section.

First, you refer to Taiwanese private hospital personnel, in line 498. However, it was not just personnel that were included as respondents in the study, as it was hospital top managers. Moreover, in line 506, you refer to senior manager of a hospital. Why? Your conclusion should focus on that group of respondents that have been included in your study.

I think that you have over-interpreted your results if you draw the conclusion that your results of this paper contribute to enhancing the success rate of hospitals in implementing cloud green e-procurement systems. You have not studied the success rate of hospitals implementing green e-procurement systems, and you have not studied the success rate of that. Therefore, you could not conclude you study in this way, as this relationship is not proven in your paper. See lines 509-512.

Neither, you can claim that the cost of investment in information systems can be reduced and efficiency of procurement activities for hospitals can be improved by using a cloud green e-procurement system, as you have not investigated that, lines 513-515.

You have to be careful just to relate and draw conclusions to what the study actually is about. It is not ok to make such speculations that you have now.

Author Response

This manuscript is a resubmission of an earlier submission. The following is a list of the peer review reports and author responses from that submission.

Round 1

Reviewer 1 Report

The authors of the paper analyse the various technological factors of adopting green e-procurement systems.

The research method using MCDM (Fuzzy Analysis Hierarchical Process) is concisely and well written, and supported by relevant literatures on sample size of experts queried. However, certain references are slightly outdated.

One query regarding the definition and difference between e-procurement used currently and e-procurement using cloud architectures (Lines 50 to 52) since both seemed to refer to the same thing?

Another query: “At present, the procurement activities of most healthcare institutions have been carried out through electronic procurement (e-procurement) system. However, the e-procurement environment based on the cloud architecture is still in the growth stage.”

SAP Ariba platform (???)

I see this paper more like a review paper format than a research article. Can the authors look into it?

Author Response

Response to Reviewer 1:
Comment #1:
The research method using MCDM (Fuzzy Analysis Hierarchical Process) is concisely and well written, and supported by relevant literatures on sample size of experts queried. However, certain references are slightly outdated.
Response #1:
Thank you.

Comment #2:
One query regarding the definition and difference between e-procurement used currently and e-procurement using cloud architectures (Lines 50 to 52) since both seemed to refer to the same thing?
Response #2: Thank you for your comment. E-procurement is an information system. It can be developed in a cloud architecture. We have rewritten this paragraph and have modified all sections related to e-procurement in a cloud architecture (please see Lines 60–64).

Comment #3:
Another query: “At present, the procurement activities of most health care institutions have been carried out through electronic procurement (e-procurement) system. However, the e-procurement environment based on the cloud architecture is still in the growth stage.” SAP Ariba platform (???) I see this paper more like a review paper format than a research article. Can the authors look into it?
Response #3:
Thank you for your comment. E-procurement may be developed by health care institutions or with the support of an independent software vendor. Software programs such as Acumatica Cloud ERP (Amazon Web Services), Spend 365, or the SAP Ariba platform can be bought or rented by companies. We have added a description (please see Lines 60–64).

Reviewer 2 Report

In this manuscript, the authors used the fuzzy analysis hierarchical process (FAHP) with technology-organization-environment (TOE) framework to understand the importance of 27 factors influencing healthcare industry adoption the green-procurement system via the cloud computing service model. The study is interesting, however, a number of issues should be addressed before its publication.

Major issues:
1. The description and explanation of Figure 1 is not clear. For example, what are the x-axis and y-axis? Explain the meaning of alpha = 1, alpha = 0.5, alpha = 0, lamda = 1, lamda = 0.5 and lamda = 0 in Figure 1.
2. Provide the questionnaires as the supporting information.

Minor issues:
1. Mathematical symbols need be improved. Superscript, subscript, lower subscript should be clearly marked. Symbols need to be consistent.
2. There are some grammar errors in manuscript and the author should make corresponding revision.

Author Response

Response to Reviewer 2:

Comment #1: The description and explanation of Figure 1 is not clear. For example, what are the x-axis and y-axis? Explain the meaning of alpha = 1, alpha = 0.5, alpha = 0, lamda = 1, lamda = 0.5 and lamda = 0 in Figure 1.
Response #1:
The Methodology section has been simplified and rewritten to clarify the procedure followed in this study. Alpha value has also been defined (please see Figure 1 and Lines 194–197).

Comment #2: Provide the questionnaires as the supporting information.
Response #2:
We thank the reviewer for their recommendation. The items in the questionnaire are provided in Table 1. Table 2 has been added to explain our questionnaire’s structure (please see Table 3 and Lines 246–252).

Comment #3: Mathematical symbols need be improved. Superscript, subscript, lower subscript should be clearly marked. Symbols need to be consistent.
Response #3:
Thank you for your suggestion. We have rechecked and revised the mathematical symbols and other symbols for consistency.

Comment #4: There are some grammar errors in manuscript and the author should make corresponding revision.
Response #4:
We appreciate your feedback and suggestions. The manuscript has been revised and proofreading services have been provided by a native speaker. Thank you.

Reviewer 3 Report

This paper is presents an analysis on influence factors of healthcare industry adopting green e-procurement system of cloud model, according its title.

The paper miss some initial explanations of the characteristics of the healthcare industry. Who is the supplier and who is the customer? Healthcare institutions are also mentioned, but the context is missing. What is then the healthcare industry? Is it an industry providing materials to the healthcare sector, or is it a provider of the healthcare? This is unclear, and therefore it is difficult to understand the essence of the whole paper.

It sounds not relevant to related the successful adoption of green e-procurement systems through the cloud computing service based in this study of factors influencing healthcare industry. May the factors can inform the healthcare industry, but just to be aware of the factors, does not imply that the factors improve the adoption. 

The introduction is not introducing the paper, as you introduce with corporates and corporate social responsibility, when healthcare and green e-procurement should be in focus, according to both the title and the abstract.

However, in the last paragraph of the introduction you write that the procurement activities of most healthcare institutions have been carried out through electronic procurement systems. So it is hard to se the relevance for this study.

What is information investment? This is mentioned in the last sentence in the introduction, and it is hard to understand why this make sense.

In the literature review you relate to a Norwegian example, why where is the example, and what does it tell? What will Schaltenbrand's reference provide to your literature review?

How does the procurement relate to supply chain management, and why do you use the concept of supply chain management in the paper? What is the contribution of this concept to the understanding of the paper?

Why is it relevant to use the TOE framework? You are also mentioned other theories, such as resource-based theory, dynamic competence theory, but why? Is there any similarities in the essence of these theories that you are using?

One line 118 you refer to and above description, but which description? Your also state that current research on hte use of cloud computing service for green procurement in the healthcare industry is lacking, but why ochandhow? Do you refer to the theories mentioned? It is difficult to understand the application of the TOE framework in this area. However, I think it is very difficult to completely examine various factors from different aspects, using the TOE framework. Has anybody had any success with that?!

There are major methodological problems in the paper. Criteria is mentioned, and that they have been reviewed, but how? Interviewees are used, but how and why? The expert discussion is vague and is repeated, and the arguments are not convincing.

The results and analysis section is speculative, and as such it is not relevant to the study at all. You also describe that the willingness of investment is low, why is this relevant? Moreover, social skills is mentions, nor is this relevant in relation to the three-layer hierarchy factors.

The paper has also some language problems. These problems make it hard to understand the whole essence of the paper in some parts.

Author Response

Response to Reviewer 3:

Comment #1
The paper miss some initial explanations of the characteristics of the health care industry. Who is the supplier and who is the customer? Health care institutions are also mentioned, but the context is missing. What is then the health care industry? Is it an industry providing materials to the health care sector, or is it a provider of the health care? This is unclear, and therefore it is difficult to understand the essence of the whole paper.
Response #1:
Thank you for your valuable comment. The problem you mentioned has been corrected. Please see Lines 40–44.

Comment #2
It sounds not relevant to related the successful adoption of green e-procurement systems through the cloud computing service based in this study of factors influencing health care industry. May the factors can inform the health care industry, but just to be aware of the factors, does not imply that the factors improve the adoption.
Response #2:
Thank you for your comment. The purpose of this paper was to enhance the success rate of health care industries adopting green e-procurement systems through optimal resource allocation according to the weight of factors and the reductions of the cost of investment in information systems. The authors have amended their statement. Please see Lines 92–104 and Lines 386–390.

Comment #3
The introduction is not introducing the paper, as you introduce with corporates and corporate social responsibility, when health care and green e-procurement should be in focus, according to both the title and the abstract.
However, in the last paragraph of the introduction you write that the procurement activities of most health care institutions have been carried out through electronic procurement systems. So it is hard to see the relevance for this study.
Response #3:
Thank you for pointing out this issue. The authors have rechecked their statement and have revised it. Please refer to the red parts in the Introduction section.

Comment #4
What is information investment? This is mentioned in the last sentence in the introduction, and it is hard to understand why this make sense.
Response #4:
Thank you for pointing out this issue. It is a grammatical error on our part. Investment in information systems is correct. Please see Line 98. The manuscript has been revised and proofreading services have been provided by a native speaker. Thank you.

Comment #5
In the literature review you relate to a Norwegian example, why where is the example, and what does it tell? What will Schaltenbrand's reference provide to your literature review?
Response #5:
Thank you for your comment. We have rewritten this sentence in the revised manuscript. Please see Lines 115–118 and Lines 120-124.

Comment #6
How does the procurement relate to supply chain management, and why do you use the concept of supply chain management in the paper? What is the contribution of this concept to the understanding of the paper?
Response #6:
We thank the reviewer for their input. We have rewritten the Introduction section. Please see Lines 53–58 and Lines 92-104. A contribution section has also been added to the Conclusion section. Please see Lines 386-390.

Comment #7
Why is it relevant to use the TOE framework? You are also mentioned other theories, such as resource-based theory, dynamic competence theory, but why? Is there any similarities in the essence of these theories that you are using?
Response #7:
We thank the reviewer for this suggestion. We have added the relevant statement regarding the use of the TOE framework. Please see Lines 81–88. Moreover, numerous studies have adopted TOE when addressing the adoption of information systems (see Lines 143–151). This is why TOE was used as the research framework in this paper.

Comment #8
One line 118 you refer to and above description, but which description? Your also state that current research on the use of cloud computing service for green procurement in the health care industry is lacking, but why ochandhow? Do you refer to the theories mentioned? It is difficult to understand the application of the TOE framework in this area. However, I think it is very difficult to completely examine various factors from different aspects, using the TOE framework. Has anybody had any success with that?!
Response #8:
We thank the reviewer for their input. The statement has been amended. No relevant studies have covered the overall factors, regardless of the theoretical framework used. This can be studied in the future, although the authors have tried to add a more precise description to address the issue mentioned. Please see Lines 81–88 and Lines 143–151. To overcome this problem, the authors have also included this in the research limitations and future study directions in the Conclusion section. Please see Lines 393–397.

Comment #9
There are major methodological problems in the paper. Criteria is mentioned, and that they have been reviewed, but how? Interviewees are used, but how and why? The expert discussion is vague and is repeated, and the arguments are not convincing.
Response #9:
Thank you for your comment. Expert questionnaire in this context actually refers to a survey questionnaire. We have amended the part regarding the criteria and the expert questionnaire survey. Please see Lines 235-240 and Lines 271–272. Moreover, the questionnaire and respondents’ answers have been provided. Please see Table 3 and Lines 251–257.

Comment #10
The results and analysis section is speculative, and as such it is not relevant to the study at all. You also describe that the willingness of investment is low, why is this relevant? Moreover, social skills is mentions, nor is this relevant in relation to the three-layer hierarchy factors. © 1996-2019 MDPI (Basel, Switzerland) unless otherwise stated.
Response #10:
We thank the reviewer for the suggestion. The research results appear to be insufficient from your perspective. However, this result has been verified using social science theories. We have not considered all possible factors or all existing theoretical frameworks, but this can be completed by other authors from other perspectives. We have also added research limitations and included this remark in the future research directions in the Conclusion section. Please see Lines 393–397.

Comment #11
The paper has also some language problems. These problems make it hard to understand the whole essence of the paper in some parts.
Response #11:
Thank you for your comment. The manuscript was revised and proofreading services were provided by a native speaker.

Round 2

Reviewer 2 Report

This manuscript is well-presented and many comments have been addressed except some.

Major issues:
1. The description and explanation of Figure 1 is still not clear. What do the x-axis and y-axis represent? Fuzzy number? Defuzzificaiton? What are the meaning of alpha = 0, 0.5, and 1?

2. It should be exemplified how the data in Table 4 is calculated. In addition to E, other A~D are calculated by E.

Minor issues:
1. Line 209, "n is the matrix array" : Is this array n the same as the number n? The authors use the same symbol in the manuscript.
2. Line 282, 346: Should be Table 4 instead of Table 3

Reviewer 3 Report

This is the second version of the paper about factors influencing healthcare industry adoption of green e-procurement using a cloud model.

The abstract refers to a priority order and weight of factors, without any relation to any factors at all. It is hard to understand how paper use, electric energy consumption etc refers to the healthcare green e-procurement, and if those are the factors the industries have to understand. Factors are also in focus related to the TOE framework, but the factors are still black-boxed. Thus, there are some problems in the logics of the abstract, that will also appear in the rest of the paper.

The first paragraph in the introduction is not including healthcare at all. Instead, supply chain management practiced by enterprises related to corporate social responsibility is highlighted. Healthcare is suddenly referred to medical care services in the second paragraph. I would like that healthcare service could have a stronger focus, also in the first paragraph, as healthcare is of vital importance in the paper.

It is unclear if the healthcare services are provided by public or private actors, or both. Healthcare refers to medical care services provided by industries, and this indicate that is a private sector. It should be clarified if the medical services are provided by a public healthcare sector or not.

What is green procurement, and what is green e-procurement? Is the use of green suppliers a requisite for green procurement? And what is green procurement activities? Are those activities required in order to adopt a green e-procurement system?

In the introduction you refer to the Australia’s public health sector, and I still wonder if you also refer to the very same sector in your paper. In line 59 you mention healthcare institutions, not industries.

In lines 51-52 you have to correct the English language.

In cannot see that it should be relevant to introduce the more technical aspects of the architecture of an e-procurement system or how the database and the system is built, as mentioned in lines 60-64, as it is not the technical details of the e-procurement system you are analyzing.

As you write about the strict requirements for green procurement in the healthcare industry, one can wonder who set the restrictions.

Moreover, the factors is mentioned here and then, without any reference to what kind of factors. Maybe you should introduce the TOE framework much earlier in your paper, as I can understand much later in the paper that the factors refer to the TOE framework. It would be fruitful if the area of interest that shape those factors could be revealed in an early phase in the paper. Also, when introducing the MCDM questionnaire, the factors are benighted. You also tell that the MCDM method was used to analyze the factors, without any hint of what kind of factors that should be analysed.

Further, in the introduction you refer to cloud computing systems in SCM that can reduce the complexity of certain existing business processes. But what is the complexity about? And what business processes in the healthcare sector will the SCM help to resolve?

You are also referring to manufacturing companies that adopted cloud environments. But how can you generalize this study to include the healthcare industries? Are there any business processes that are similar to those within the healthcare industries? Is efficiency of purchasing and the status of purchasing raw material of vital importance in the healthcare industries in your practical setting?

Sometimes you use the concept of green procurement and sometime e-procurement, and it is not always clear what those concepts stands for.

Why is the purpose of the paper to analyze the attitude of those in healthcare industries adopting green e-procurement through cloud computing? You have not fully introduced this to the reader.

Thus, some of the main problems in the paper refers to the introduction.

In line 138 you write about the challenge of a high level of uncertainty. Why is there a high level of uncertainty in the current green procurement environment? And how about the green e-procurement environment?

You need references for the lines 154-158.

Also in the research method you refer to the factors, without actually mention what types of factors you will evaluate.

The result and analysis section seems to include speculations, for example as you refer to most doctors in lines 285-286.

It is also questionable if the study results infer your statements in lines 298-301. Also the paragraph in lines 346-360 seems to be speculations, as I cannot see that statements can be derived from the results.

In conclusions you mention adaptation of systems, is this correct?

You should not have references in the conclusions section.

What are actually your conclusions of the study? What factors are specific for the green e-procurement system in this healthcare setting?